# Nutrient and Chemical Contaminant Levels in Five Marine Fish Species from Angola—The EAF-Nansen Programme

**DOI:** 10.3390/foods9050629

**Published:** 2020-05-14

**Authors:** Amalie Moxness Reksten, Avelina M. Joao Correia Victor, Edia Baptista Nascimento Neves, Sofie Myhre Christiansen, Molly Ahern, Abimbola Uzomah, Anne-Katrine Lundebye, Jeppe Kolding, Marian Kjellevold

**Affiliations:** 1Seafood, Nutrition and Environmental State, Institute of Marine Research, P.O. Box 2029, Nordnes, 5817 Bergen, Norway; sofie.christiansen22@gmail.com (S.M.C.); anne-katrine.lundebye@hi.no (A.-K.L.); marian.kjellevold@hi.no (M.K.); 2Quality Control Department of Fisheries Products, National Institute of Fisheries and Marine Research, P.O. Box 2901, Luanda, Angola; avelinajoao22@gmail.com (A.M.J.C.V.); elizabeth23102005@hotmail.com (E.B.N.N.); 3Fisheries and Aquaculture Division, Food and Agriculture Organization of the United Nations (FAO), 00153 Rome, Italy; molly.ahern@fao.org; 4Department of Food Science and Technology, Federal University of Technology, P.M.B. 1526, Owerri 460114, Nigeria; abimuzomah@yahoo.com; 5Department of Biological Sciences, University of Bergen, P.O. Box 7803, 5020 Bergen, Norway; jeppe.kolding@uib.no

**Keywords:** nutrients, fish, contaminants, Angola, metals, marine, recommended nutrient intake, exposure assessment, minerals, food composition data

## Abstract

Fish is a rich source of several important nutrients and an important part of the otherwise plant-dominated diet present in Angola. However, fish may also be a source of contaminants. The aim of this study was to analyse the nutrient contents and the levels of chemical contaminants, including arsenic, cadmium, mercury, and lead, in five commonly consumed marine fish species sampled during a survey with the research vessel *Dr. Fridtjof Nansen* in Angola. The species’ contribution to recommended nutrient intakes (RNI) for women and children was assessed and compared to that of food products of terrestrial animal origin. All the sampled species are good sources of protein and micronutrients if included in the diet, and inter-species variation is evident. The species were identified to contribute 5–15% of the RNI for calcium, iron, iodine, and zinc and exceeded the contribution to protein and iron intakes of food products of terrestrial animal origin. Furthermore, the potential consumer exposure to chemical contaminants in the species was assessed. None of the species exceeded the maximum levels for cadmium, mercury, and lead, and the potential consumer exposure to cadmium and methylmercury was considered low. The data presented in this study represent an important contribution to African food composition tables.

## 1. Introduction

With a 1650 km long coastline located in one of the most productive ecosystems in the world [1], Angola’s fishery sector is very important to the country’s economy and represents a valuable source of food and livelihood to a large part of the population, particularly in the coastal areas. Following the oil and diamond industries, fisheries represent the third most important economic sector. Of the 486,490 recorded tonnes landed annually from capture fisheries, marine resources account for more than 70% of the total Angolan fish production [2,3]. However, due to a tendency for misreported livelihoods and economic contributions from capture fisheries globally (hidden harvests), the production number is expected to be considerably higher [4]. Small pelagic species, which are especially important for domestic food supply, represent approximately 50% of the total reported catch, with sardinella (*Sardinella* spp.) and horse mackerel (*Trachurus trachurus*) being the most abundant and preferred species. Other common marine fish resources include demersal finfish such as sea breams, snappers, groupers, and croakers and migratory species of tuna. An estimated 90% of Angolan fish production is marketed within national borders, and most of the fish is sold and utilised fresh or frozen at local landing sites and the rest as either salt-dried or sun-dried fish. Estimates from 2016 indicate a per capita annual fish consumption of 19.5 kg [5], which translates to approximately 26% of the total animal protein intake [2]. This compares against a world per capita estimated average consumption of 19.7 kg and an African average of only 9.9 kg per capita [5]. 

After a 27-year long civil war, which came to an end in 2002, Angola’s public health system is still in the process of being rebuilt. Consequently, health data and information on the nutritional status of the Angolan population are scarce and scattered [6]. Food insecurity and malnutrition remain great challenges, with iron, iodine, vitamin A, and zinc deficiencies as the most prevalent micronutrient deficiencies [7]. In sub-Saharan Africa, the prevalence of iron deficiency in women is the highest across the globe [8]; estimates from 2016 show that 48% of women of reproductive age and 51% of pregnant women in Angola suffer from anaemia (of both nutritional and non-nutritional aetiologies) [9]. Furthermore, the mortality rate for Angolan children under five years of age is the sixteenth highest in the world [10], and almost half of all child deaths are attributed to poor nutrition. It has been estimated that 87% of children between 6 and 23 months old do not receive a minimum acceptable diet [11], with a prevalence of 38% of under-five year olds classified as having stunted growth, which is considerably higher than the developing country average of 25% [12]. Additionally, critical levels of food and nutrition insecurity exist due to recurrent disease outbreaks, drought, and flooding incidents [11], which has led to food and nutrition security being identified as one of ten priorities in Angola’s national strategy to fight poverty [3].

As an important part of the African agri-food system, fisheries are seen as a crucial contributor to combating hunger and acquiring food security, both directly through consumption and indirectly through providing work and income-generating opportunities [13]. Fish is widely recognised as a rich source of high-quality protein, long-chain omega-3 fatty acids such as eicosapentaenoic acid (EPA) and docosahexaenoic acid (DHA), various vitamins (A, D, and B_12_), and a range of minerals (including calcium, iron, zinc, iodine, and selenium) [5,14,15]. Several of these nutrients cannot be easily substituted by other food commodities [8,16]. However, fish may also be a source of chemical contaminants including mercury, lead, cadmium, and arsenic, either through naturally occurring phenomena such as volcanic eruptions and weathering and/or as a result of anthropogenic activities such as industrial processes and agricultural activities [17,18,19]. These metals and metalloids (hereafter combined and referred to as “metals”) have raised serious public health concerns over the past decades, as exposure is associated with a range of negative health effects, including renal, skeletal, cardiovascular, neurological, and developmental damage in humans [20,21,22]. 

Adequate nutrition has been recognised as an essential catalyst for achieving the United Nation’s Sustainable Development Goals (SDGs) by 2030 [23]. However, as highlighted in the 2017 Global Nutrition Report [24], major data gaps in the nutrient composition of foods consumed in Africa, and thus accurate dietary data on current nutrient intakes, have been identified as hindrances to development and progress in African countries. Reliable and up-to-date knowledge on the composition of foods, or food composition data, represent a vital tool and source of information for the development of food policies and the implementation of intervention programmes to improve health and food and nutrition security in all populations [24,25]. Along with numerous other African countries, Angola does not currently have its own food composition table/database (FCT/FCDB). A few sub-Saharan countries, including Cameroon, Congo, Zambia, and Zimbabwe, have outdated FCTs from 1957–1989, whereas South Africa is one of very few African countries to have a more recently updated FCT/FCDB (from 2008, last updated in 2017) [24,26]. On a regional level, the Food and Agriculture Organization of the United Nations (FAO)/International Network of Food Data Systems (INFOODS) Food Composition Table for Western Africa (2019), which was released in 2020 and replaced the former version from 2012 [27,28], is the most relevant FCT for use in Angola, as no regional FCT currently exists for countries in central Africa. The new FCT includes 1028 food entries (compared to 472 in the former FCT from 2012), several mixed dishes (mixed dishes were not included in the former FCT), 29 new components/nutrients (in addition to the 28 present in the former FCT), and 300 new data sources (raising the number from 121 in the former FCT to 467 in the new version). For the food group “Fish and its products”, the new FCT contains nutrient data for 21 (raw) fish species, which is an increase from the 15 fish species described in the FCT from 2012. Additionally, the FCT contains calculated values for canned, boiled, grilled, and steamed fish [27,28,29]. As stated in the FAO/World Health Organization (WHO) expert consultation on the risks and benefits of fish consumption [30], countries are recommended to develop, maintain, and improve existing databases on nutrients and contaminants in fish consumed in their region. Thus, the aim of this study was to quantify a variety of nutrients and chemical contaminants in commonly consumed marine fish species sampled from the coast of Angola in order to fill current knowledge gaps on the composition of fresh fish and contribute to high-quality food composition data. Furthermore, we assessed the potential contribution of calcium, iodine, iron, and zinc to the recommended nutrient intakes (RNI) for women of reproductive age and infants during the first 1000 days of life; groups that are both considered nutritionally vulnerable and where adequate nutrition is one of the fundamental prerequisites for optimal development and health [31]. We also estimated the potential human exposure to cadmium and methylmercury (MeHg) for adults and young children when consuming the sampled species. 

## 2. Materials and Methods 

This paper uses data collected by a scientific survey with the research vessel (R/V) *Dr. Fridtjof Nansen* as part of the collaboration between the EAF-Nansen Programme and the National Fisheries Research Institute (INIP) in Angola. The EAF-Nansen Programme is a partnership between the FAO, the Norwegian Agency for Development Cooperation (Norad), and the Institute of Marine Research (IMR), Norway, for the sustainable management of the fisheries of partner countries.

### 2.1. Sampling

Sampling of marine fish species off the coast of Angola was performed on board R/V *Dr. Fridtjof Nansen* between 21 September and 12 October 2017. Pelagic (Åkrehamn) and demersal trawls (Gisund Super bottom trawl) were used for sampling during the survey, and the catch was subsequently sorted and identified according to species by taxonomists. Fish species were selected for nutritional analyses based on their importance to local food habits as advised by the local marine and food scientists on board. The fish were measured (from the tip of the head to the deepest fork of the caudal fin) and weighed on a marine measuring board, and average weight (g) and length (cm) were recorded for each species (Table 1). For each fish species, a total of 25 individuals were filleted (including removal of the head, tail, skin, bones, and viscera of the fish; fillet samples from both sides of the fish were included from each fish sample) and homogenised using a food processor (Braun Multiquick 7 K3000, Kronberg im Taunus, Germany). The 25 fillet samples were then pooled together, creating five composite samples consisting of fillets of five individuals each (5 × 5) that were homogenised once more in the same food processor. The composite samples were stored as wet samples for later analyses at −20 °C in the vessel’s freezer. After a minimum of 12 h in the freezer, a subsample of each wet sample was freeze-dried for 72 h (24 h at −50 °C, immediately followed by 48 h at +25 °C, with a vacuum of 0.2–0.01 mbar), using a Labconco FreeZone 18 L freeze-dryer (Kansas City, mod. 7750306, MO, USA), and the moisture content was calculated based on the weight change between entering and exiting the freeze-dryer. Freeze-dried samples were then homogenised using a knife mill (Retch Grindomix GM 200, Haan, Germany). All samples were vacuum-sealed and stored in airtight plastic beakers (Nunc beakers) in insulated boxes in the freezer (−20 °C) until shipment by air cargo to the IMR in Bergen, Norway, where the samples were stored at −80 °C pending analyses. The samples arrived at the IMR laboratories in early January 2018. Detailed information regarding the sampling procedures is described in Reksten et al. [32]. 

### 2.2. Analytical Methods

The determination of crude protein, lipid content, minerals, and chemical contaminants was performed in singular parallels at the IMR laboratories in Bergen, Norway. Details of the analytical methods are given by Reksten et al. [32].

Lipids (crude fat) were extracted with ethyl acetate and filtered before the solvent was evaporated and the lipid residue was weighed. The method is standardised as a Norwegian Standard, NS 9402 [34]. Protein (crude protein) was determined by burning the material in pure oxygen gas in a combustion tube at 950 °C. Nitrogen (N) was detected with a thermal conductivity detector, and the content of N was calculated from an estimated average of 16% N per 100 g protein. The following formula was used: N g/100 g × 6.25 = g protein/100 g, in accordance with the method accredited by the AOAC Official Methods of Analysis [35].

The concentrations of minerals (selenium (Se), zinc (Zn), iron (Fe), calcium (Ca), potassium (K), magnesium (Mg), phosphorus (P), and sodium (Na)) and metals (arsenic (As), cadmium (Cd), mercury (Hg), and lead (Pb)) were determined with an inductively coupled plasma-mass spectrometer (ICP-MS, iCapQ ICPMS, ThermoFisher Scientific, Waltham, MA, USA) equipped with an auto-sampler (FAST SC-4Q DX, Elemental Scientific, Omaha, NE, USA) after wet digestion in a microwave oven (UltraWave, Milestone, Sorisole, Italy), as described by Julshamn et al. [36]. The concentrations of these minerals were quantified using an external standard curve in addition to an internal standard [37]. Three slightly different methods were applied: (1) for Ca, Na, K, Mg, and P, using scandium (Sc) as the internal standard; (2) for Zn and Se, using rhodium (Rh) as the internal standard; and (3) for iodine (I), tellurium (Te) was used as the internal standard. For the determination of I, the sample preparation was a basic extraction with tetramethylammonium hydroxide (TMAH) before ICP-MS analysis.

### 2.3. Data Management and Presentation of Analytical Data

The analytical data were exported from the Laboratory Information Management System (LIMS) and plotted into spreadsheets in Microsoft^®^ Office 365 Excel version 1910 for the calculation of means and standard deviations (SD). The data presented are reported with the same units of expression and rounding procedures as advised in the FAO guidelines for food composition data [25]. Statistical analyses were performed, and graphs compiled using GraphPad Prism version 8.3.0, 2019. The data did meet the normality assumption (tested using the Shapiro–Wilk normality test); thus, Pearson’s correlation coefficient was used to test the correlation between moisture and lipid content. The analytical values for protein, lipids, and minerals are presented as the mean ± SD per 100 g wet weight (w.w.) for the five composite samples from each fish species. For metals, the values are presented as the mean ± SD expressed in mg/kg w.w. of the five composite samples for each species. No values were <LOQ for any of the nutrients analysed, but for individual samples presenting values <LOQ for any of the chemical contaminants, we applied a conservative approach assuming that the total amount of the contaminant present in the sample is equivalent to the LOQ value. For arsenic and mercury, no values were <LOQ. For cadmium, six of 30 measurements were <LOQ, and for lead, 25 of 30 measurements were <LOQ. The concentration of mercury was measured as total mercury in this study; it can be assumed that 80–100% of the total mercury in fish is in the form of MeHg [38]. Similarly, the concentration of arsenic was measured as the total concentration of all arsenic compounds present in the samples.

### 2.4. Calculation of Potential Contribution to Recommended Nutrient Intakes

The potential contribution of each species to the daily RNI was calculated with reference to recommendations for non-pregnant, non-lactating, healthy females of reproductive age (aged 19–50 years) and for infants during the first 1000 days of life. The average RNI values for infants were calculated to account for variations in requirements throughout the age period from 7 to 23 months [39], assuming exclusive breastfeeding for the first six months. The micronutrients of interest were calcium, iodine, iron, and zinc. Despite a lack of national data on the composition of the Angolan diet, low dietary bioavailability was assumed for both iron (10%) and zinc (15%) based on dietary trends observed in neighbouring African countries, including a limited intake of animal-source food products, few sources of vitamin C-containing foods, and a high consumption of phytate-containing foods such as cereals, roots, and legumes [40,41]. The potential contribution from each fish species was calculated using a 50 g portion of raw fish for women (estimated from the most recent per capita fish consumption of 19.5 kg/year [5]) and an estimated 25 g portion of raw fish for children 7–23 months of age. The values for each species are presented as percentages of the average RNI for each group.

### 2.5. Comparison with Food Products of Terrestrial Animal Origin

The various species’ potential contribution to the RNI of an average African woman weighing 60 kg was compared to that of raw chicken, beef, pork, milk, and eggs with regard to protein, calcium, iron, and zinc concentrations. For protein, a dietary protein requirement of 0.84 g per kg body weight per day, as advised by the FAO/WHO [42], was utilised. The nutrient values for each food product of terrestrial animal origin were derived from the FAO/INFOODS Food Composition Table for Western Africa (2019) [28]. A description of the food products as well as the edible portion conversion factors used for the calculation of each food’s potential contribution are presented in Table 2. All values were compared in their raw state, using an equivalent portion size of 50 g for both fish and food products of terrestrial animal origin.

### 2.6. Potential Consumer Exposure to Metals

An evaluation of the potential consumer exposure to cadmium and mercury through the consumption of the sampled species was performed by evaluating the estimated daily intake (g) compared to the provisional tolerable weekly/monthly intake (PTWI/PTMI) of the two metals, as provided by the Joint FAO/WHO Joint Expert Committee on Food Additives (JECFA). THE PTWI is defined as the maximum intake of contaminants in food that may be consumed on a weekly basis over a lifetime without causing any adverse health effects. For MeHg, the PTWI is set at 1.6 µg/kg body weight [43], whereas for cadmium, the JECFA has determined that indicating the tolerable intake monthly is more appropriate than weekly due to the metal’s long half-life. Thus, the PTMI for cadmium is set at 25 µg/kg body weight [44]. For the exposure assessment of mercury, a precautionary approach was applied assuming that the total mercury in fish was entirely in the form of MeHg [38]. As a worst-case scenario for cadmium, the consumer exposure was calculated assuming a fish intake of 30 times per month (one portion/day) for each fish species. The calculations were not performed for arsenic and lead as no PTWI values are available for these metals. In this paper, the potential consumer exposure was assessed for an average African adult of 60 kg and a 23-month-old child of 12 kg, as depicted in the WHO growth chart for boys (the 50th percentile was selected) [45,46]. Fish serving sizes equivalent to the ones utilised in the calculations for RNI (50 g fish/day for adults and 25 g fish/day for infants) were used when calculating the potential consumer exposure.

## 3. Results and Discussion

This paper is the first to present comprehensive analytical information on the nutrient composition and concentration of metals in five commonly consumed marine fish species sampled off the coast of Angola. The nutrient composition of two of the sampled species, *Lagocephalus laevigatus* (described in the Appendix A due to its low relevance as a food fish, Appendix A) and *Trachurus trecae*, has never been reported in the scientific literature before. To the best of our knowledge, this is also the first study to analyse and report on the mineral content of *Caranx rhonchus* and *Sardinella maderensis* and the first study to report the arsenic, cadmium, mercury, and lead content in any fish species sampled from Angola.

### 3.1. Proximate Composition

The proximate composition of the sampled fish species is described in Table 3. The protein content ranged from 19 to 23 g/100 g with the highest content identified in *Caranx rhonchus*. In a previous study from Tunisia, this species was identified as being particularly high in muscle protein content with values ranging from 18 to 22 g/100 g when compared to other marine fish species [47]. The protein content in fish is fairly stable, typically varying between 15% and 20% in muscle tissue, and is considered to be of high quality due to the balanced amino acid profile consisting of significant amounts of all nine essential amino acids for human nutrition [48,49,50,51]. Additionally, the protein found in fish is easily digestible due to there being very little connective tissue [52,53]. Previous studies have found that even small amounts of high-quality protein from animal sources such as fish yield significant improvements in maternal health and child development when included in plant-based diets due to a high content of micronutrients and other enhancing properties (referred to as the “meat factor”) [54,55]. The composition of lipids in fish is the most heterogeneous component and may differ according to geographical region, seasonal variations, environment (water temperature, salinity, and pressure), diet/food supply, and the maturity, sex, and reproductive stage of the fish [48,51,56,57]. In this paper, the total lipid content varied greatly with a range of 0.43–6.98 g/100 g, categorising all fish species (with the exception of *Lagocephalus laevigatus*) as intermediate (2–8% of the total lipid content) according to the Norwegian and Danish categorisation of lipid content as a percentage of total body weight [58]. As previously described by others, and as shown by a negative correlation coefficient of 0.8 in this paper, an inverse relationship exists between the water content and the lipid content in fish where the water content increases as the content of lipids decreases [48,58].

### 3.2. Mineral Composition

The concentrations of selected minerals in the five sampled fish species are presented in Table 4. Overall, the mineral content varied considerably among the fish species. When comparing the mineral levels to the FAO mean concentration range in fish muscle as compiled by Mogobe et al. [59], all values fall within the mean ranges apart from those for potassium and iron. The potassium content in the sampled species (456–596 mg/100 g) is slightly higher than the FAO mean range of 19–502 mg/100 g, whereas the iron range is marginally lower (0.23–1.84 and 1–5.6 mg/100 g, respectively). As addressed in several other studies [16,60,61,62,63,64], which anatomical components of the fish are analysed and subsequently consumed (bones, skin, head, viscera) significantly affects the nutrient yield. For example, in a study from Cambodia, Roos et al. [64] reported a 60% reduction in total iron content in the small indigenous species *E. longimanus* when the head and viscera of the fish were removed, suggesting that this may be where most of the iron in fish is stored. Similarly, approximately 99% of the accumulated calcium and 80% of the phosphorus is stored in the bones, teeth, and scales of fish [57,65]. This explains the low calcium levels found in the sampled species from Angola compared to in other studies where whole fish have been analysed [62,63,66], as neither heads nor viscera were included in the analyses in this paper. In a study including 367 species of fish from 43 different countries, Hicks et al. [15] reported higher concentrations of calcium, iron, and zinc in species from warmer and shallower waters compared to in polar and/or deep-water species. If comparing the clustered means for zinc and iron in the sampled species from Angola to mean values in fillets of five of the most commonly consumed marine wild fish species in the North Atlantic sea (haddock, Atlantic cod, Atlantic halibut, Atlantic mackerel, and Atlantic salmon), the zinc content of the sampled species from Angola (0.46 ± 0.1 mg/100 g) exceeded that of haddock, cod, halibut, and salmon (0.3 ± 0 mg/100 g, 0.4 ± 0.1 mg/100 g, 0.43 ± 0.2 mg/100 g, and 0.4 ± 0.04, respectively). For iron, the mean content in the sampled species (1.0 ± 0.6 mg/100 g) also exceeded that of haddock, cod, halibut, and salmon (0.1 ± 0.04 mg/100 g, 0.2 ± 0.04 mg/100 g, 0.1 ± 0.1 mg/100 g, and 0.48 ± 0.2 mg/100 g, respectively) [67]. As argued by Hicks et al. [15], a plausible explanation for this difference in mineral content between various tropical climates may be the heavy rainfall that commonly occurs in tropical regions, where the micronutrients normally present in the soil are washed into the ocean and thus enter the marine food chains.

### 3.3. Contribution to Recommended Nutrient Intakes

Figure 1 and Figure 2 display the sampled fish species’ potential contributions to the RNI of women and infants age 7–23 months when portions of 50 g and 25 g, respectively, are consumed. Several species were identified to potentially contribute between 5% and 15% of the RNI for both women and infants of the selected nutrients. The potential contribution to the RNI of iodine was greater than that for the other minerals, with a mean for all species of 9.4% for women and 7.8% for infants, whereas the contribution to the RNI of calcium was the lowest with a mean value of 2% for both women and infants. Compared to other studies where the contents of similar nutrients in fish have been evaluated in reference to the RNI using a 50 g serving size, the findings of this study were less substantial. For example, when analysing the nutrient content of 55 fish species from Bangladesh, Bogard et al. [60] identified 14 and 18 species (prepared as both fillets and as whole fish) that would meet ≥50% of the RNI for calcium for pregnant and lactating women and infants (25 g serving size for infants), respectively. For iron, three species were identified to potentially meet ≥25% of the RNI for both groups (all prepared whole). In India, Mohanty et al. [68] identified several marine and freshwater species that met ≥100% of the RNI for an average man for calcium, iron, and zinc (a 25 g serving size was used when estimating the RNI of calcium). However, it is important to emphasise that different anatomical parts of the fish were included in these analyses and calculations, and as previously mentioned, which parts of the fish are consumed will significantly affect the nutrient yield [16,60,61,62,63,64]. In one of the very few African studies to describe the nutrient content in fish compared to the RNI, the mineral compositions of five freshwater fish species from Botswana have been described [59]. In this study, the authors reported considerable differences between small species consumed whole and large species that were filleted; the two small species were able to meet ≥100% of the RNI of calcium, whereas the large species were able to meet 40–60% (estimated for women, using a portion size of 100 g raw fish). The two small species were found to meet 27% of the RNI for iron and approximately 100% for zinc, whereas the large species were found to contribute considerably less. Nevertheless, in Bangladesh, Nordhagen et al. [69] reported mineral levels in several marine fish species where only the fillet (including the skin) was analysed and found that they met ≥50% of the RNI for women for calcium and ≥25% for zinc (when a 100 g portion is consumed). For iron, the values (10–15% of RNI) were in line with the results reported in this paper.

### 3.4. Comparison to Food Products of Terrestrial Animal Origin

The sampled fish species’ contribution to the RNI of protein, calcium, iron, and zinc compared to food products of terrestrial animal origin are presented in Figure 3. The sampled fish species were found to contribute more to the protein requirement with a mean protein value of 21 ± 1.3 g/100 g, compared to 15 ± 7.2 g/100 g for the terrestrial animal food products. Of the terrestrial animal food products, beef was the only product equivalent to the sampled fish species with a potential contribution of approximately 20% of the daily protein requirement of an adult. Milk and eggs were, together with the two *Sardinella* spp., the most significant sources of calcium, but a single serving was only estimated to meet approximately 5% of the RNI for women. For iron, most of the sampled fish species contributed considerably more to the RNI than any of the terrestrial animal food products. The two *Sardinella* spp. were found to both contribute between 10% and 15% to the RNI, whereas beef, the animal product with the highest iron content, only covered approximately 2.5% of the RNI. For zinc, pork and beef substantially exceeded all the fish species’ potentials to meet the RNI. Nevertheless, common to both fish and terrestrial animal food products is that they provide multiple micronutrients simultaneously, which is important in diets lacking in more than one nutrient. Furthermore, despite a relatively low contribution of the sampled fish species to RNI when compared with other studies, this comparison to terrestrial animal food products highlights the nutrient density of fish, as most of the sampled fish species are equally as nutrient dense as, or have a higher nutrient density than the selected terrestrial animal food products.

### 3.5. Fish for Food and Nutrition Security in Angola

The Angola Nutrition Gap Analysis from 2011 identified several priority interventions that should be included in national nutrition policies for Angola: micronutrient powders for young children, iron fortification of staple foods, zinc supplementation for the treatment of diarrhoea in children, and iodine supplementation for pregnant women. However, as of 2019, the interventions are yet to be finalised for inclusion in a national nutrition policy or any other policy documents [7]. Food-based strategies, including dietary diversification, is recognised as one of the most long-term sustainable and cost-effective solutions to improve food and nutrition security in low- and middle-income countries, such as Angola [70]. As demonstrated in this paper, fish is a rich source of several micronutrients, but inter-species variation is evident. The availability of accurate and up-to-date food composition data is, therefore, essential to be able to identify foods rich in micronutrients and successfully promote and implement strategies to increase the consumption of these. Considering the current low intakes of iron and zinc in developing countries, particularly by vulnerable groups like women and children [71,72], fish has the potential to make an important contribution to the intake of these nutrients. A large proportion of the iron present in fish is in the form of haem iron, which has a higher bioavailability than the non-haem iron typically found in non-animal food products such as legumes, cereals, fruits, and vegetables, including the elemental iron present in supplements and used for food fortification [16,73,74,75]. In plant-based diets such as the maize-dependent diet prevalent in Angola, the type and quality of the dietary iron consumed may therefore be of greater importance than the total dietary iron intake [76]. Additionally, muscle tissue from animal food products such as fish enhances the absorption of iron and zinc from cereal and tuber-based diets; thus, including even small quantities of fish in the diet may enhance overall micronutrient bioavailability [16,39,75,77,78]. This is of particular importance in Angola and other sub-Saharan countries, where the prevalence of iron deficiency in women is among the highest in the world [8]. In the new version of the FAO/INFOODS Food Composition Table for Western Africa (2019), three of the fish species sampled in this study are included, although referred to only by genus and not by specific species [28]. For example, for the food item “sardines”, all species of the genus *Sardinella* are included in the description (the genus includes 22 known species), and for “barracuda”, all species of the genus *Sphyraena* are included (the genus includes 28 known species). However, Atlantic horse mackerel is described as an individual food item and is specific only to the fish species *Trachurus trachurus*, not the entire *Trachurus* genus (which then would have included our sample of *Trachurus trecae*). Considerable differences between species of the same genus have been observed [69,79]. This can also be illustrated when comparing the mineral content of *Sphyraena guachancho,* as sampled in this paper, and samples of *Sphyraena jello* from Sri Lankan waters [63] (both known as barracudas); the concentrations of the minerals calcium, iron, iodine, and selenium vary considerably between the two species. A considerable difference between the two samples of *Sardinella* was also observed in this paper, where the content of particularly iodine and selenium varied substantially. The analytical data presented in this paper may therefore be of great value for the further development of the West African food composition table, either as a means to increase the number of data points/sources for already established food groups/items or as a foundation for increasing the number of species included in the FCT.

### 3.6. Concentration of Chemical Contaminants

The arsenic, cadmium, mercury, and lead concentrations in the various fish samples from Angola, expressed on a w.w. basis, are presented in Table 5. None of the samples exceeded the Europeans Union’s (EU) maximum levels for mercury, cadmium, or lead in fish (0.5, 0.050, and 0.30 mg/kg, respectively [80]). No maximum level for arsenic in fish has yet been established. Fish is a major source of arsenic exposure for humans, but the majority of the arsenic is in the non-toxic form of inorganic arsenic [81]. The total mercury concentration in the muscle tissue (*n* = 1) of *Caranx rhonchus* was found to be 1.701 mg/kg w.w. in a study from Italy [82], which is substantially higher than the results presented in our study for this species. A study from Senegal reported levels of lead in samples of *Sardinella aurita* (*n* = 50) similar to our results, whereas the level of cadmium was considerably higher (0.01–0.20 mg/kg w.w.) [83]. As this is, to the best of our knowledge, the first study to quantify the arsenic, cadmium, mercury, and lead contents in any fish species sampled from Angola’s marine waters, more studies are needed to be able to compare contaminant levels both within and across fish species from Angolan waters and to further monitor possible changes in the concentrations of metals. Moreover, in order to conduct more accurate risk–benefit analyses connected to the consumption of fish in Angola, more studies are also needed on the levels of metals and other contaminants in processed fish. The most common processing methods include salt-drying, sun-drying, and smoking, and due to the absence of a nationwide distribution system for fresh fish, it can be assumed that a large proportion of the fish is processed in some such way before transportation to nearby villages and larger towns [2].

### 3.7. Potential Consumer Exposure

Figure 4 illustrates to what extent one portion of the various fish species from Angola contributes to the PTWI and PTMI for MeHg and cadmium, respectively, for adults. The largest contribution to the PTWI for MeHg was from *Sphyraena guachancho*, which contributed 9%, whereas the lowest contribution was from *Sardinella aurita*, with a little under 2%. This means that in a worst-case scenario where *Sphyraena guachancho* is consumed every day for a week, the PTWI would still not be exceeded. A worst-case scenario was also selected for the evaluation of the consumer exposure to cadmium, where we assumed that each species is consumed every day for a month (30 times/month). We found that none of the species exceeded 1% of the PTMI; thus, it can be assumed that the sampled species from Angola are not significant sources of cadmium exposure for humans. It is also very unlikely that an individual’s entire monthly fish intake would consist of a single species. Figure 5 illustrates the potential consumer exposure to MeHg and cadmium for children. For this group, *Sphyraena guachancho* was found to contribute 22% to the PTWI of MeHg. Thus, in a worst-case scenario where this species is consumed every day for a week, the PTWI would be exceeded at 154%. However, the other species were found to contribute far less; *Sardinella maderensis* was the second largest contributor at only 6%. As the samples of *Sphyraena guachancho* were almost twice as large as the rest of the species, this observation is not unexpected. Generally, higher concentrations of mercury are observed in larger and older fish positioned higher in the food chain [38]. Mercury exposure is of particular concern in young children due to the neurodevelopmental toxicity of MeHg [84]. In this study, a conservative approach assuming that 100% of the mercury present in the sampled fish species was in the toxic form of MeHg was taken. This is in line with the approach taken in the exposure assessment for mercury conducted by the European Food Safety Authority (EFSA, [38]) and other studies where the content of MeHg has been found to account for 80–100% of the total mercury in fish. [85]. Our results showed that if *Sardinella maderensis*, the sampled fish species with the highest content of cadmium, was consumed every day for a month, this would only account for approximately 1.5% of the PTMI for children. Nevertheless, it should be mentioned that in this evaluation, only exposure to fish has been assessed. Other seafood, such as cephalopods, shrimps, lobsters, and crabs, likely constitute an important part of the Angolan diet as well, and as major sources of metal exposure [38,86], the consumer exposure to cadmium and MeHg is possibly increased when evaluating the group “seafood” as a whole.

### 3.8. Limitations

The contribution of fish to the RNI is not only determined by the nutrient content of the species, but also by the local processing methods and eating patterns [16]. Thus, studies are also needed to indicate the nutrient content of the edible parts of the fish by reflecting the local methods used to clean and prepare the fish for a meal. The fish samples in this study were filleted, but as in other African countries, it could be assumed that a large share of the fish consumed in Angola is consumed whole, especially fish of smaller size [16,31,59]. However, no dietary studies on fish intake and preparation and processing methods in Angola are available in the scientific literature. Moreover, the actual portion size consumed in the country is of great importance when estimating the RNI. As no dietary surveys or household data were found for Angola, estimating accurate and representable portion sizes for different population groups is difficult. In Nigeria, another west African coastal country, household data indicate fish intake to be between 124 and 217 g/day [87], whereas the national statistics indicate a consumption level of 24.6 g/day [16]. As argued by Kawarazuka et al. [16], scattered intake data for fish in the literature suggest a mean intake of approximately 70–150 g/day in Africa; a range where each extremity would have significantly affected the calculations of the species’ contribution to the RNI. Furthermore, the RNI in this study was calculated for raw fish; however, dietary bioavailability, the quantity of nutrients in the fish samples, and the weight and volume of the fish is influenced by the method of preparation/processing (sun-drying, salting, boiling, smoking, etc.) [25,72].

## 4. Conclusions

The levels of several nutrients and chemical contaminants in five marine fish species sampled from the coast of Angola are described in this paper. All the sampled fish species are good sources of several key nutrients when included in the diet. The species were identified to contribute 5–15% of the RNI for calcium, iodine, iron, and zinc for women and children. Compared to other studies where the whole fish (including bones, head, and viscera) were commonly consumed and included in the calculations, this is not a substantial contribution. However, when compared to the nutrient values of other food products of terrestrial animal origin—such as beef, chicken, and milk—the sampled fish species were identified to contribute more to the recommended intakes for both protein and iron, approximately the same for calcium, and less for zinc. None of the sampled fish species exceeded the EU maximum levels for cadmium, mercury, or lead, and the potential consumer exposure to cadmium and MeHg may be considered small for both adults and children. To the best of our knowledge, this is the first study where fish from the coastal waters of Angola have been analysed for nutrients and/or contaminants, and these data provide an important contribution to addressing the food composition data challenge in several African countries.

## Figures and Tables

**Figure 1 foods-09-00629-f001:**
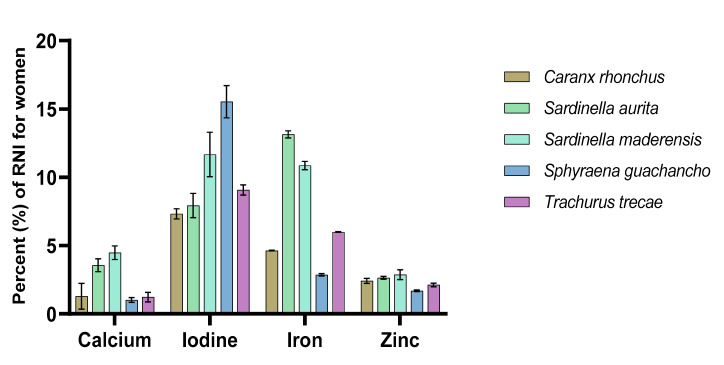
The contribution of calcium, iodine, iron, and zinc (in percentage) to the recommended nutrient intake (RNI) for women of reproductive age, from the consumption of a 50 g portion fillet of the various fish species.

**Figure 2 foods-09-00629-f002:**
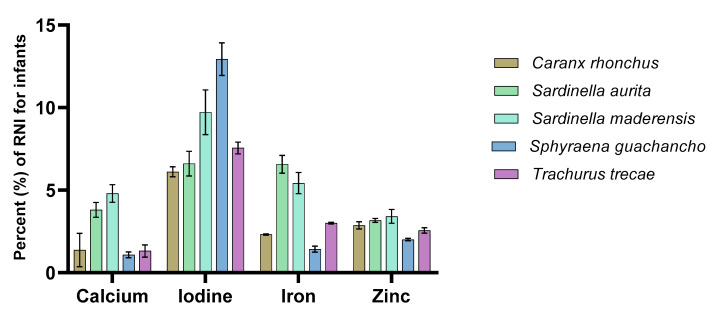
The contribution of calcium, iodine, iron, and zinc (in percentage) to the recommended nutrient intake (RNI) for infants during the first 1000 days of life, from the consumption of a 25 g portion fillet of the various fish species.

**Figure 3 foods-09-00629-f003:**
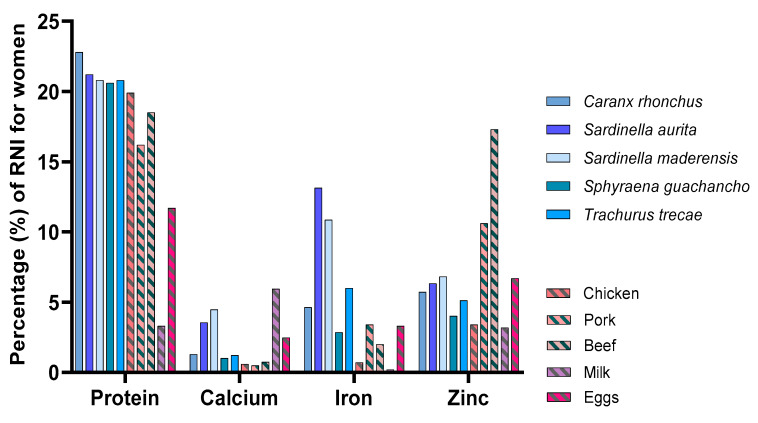
The contribution of protein, calcium, iron, and zinc (in percentage) to the recommended nutrient intakes (RNI) for women, from the consumption of a 50 g portion fillet of the sampled fish species and various terrestrial animal food products.

**Figure 4 foods-09-00629-f004:**
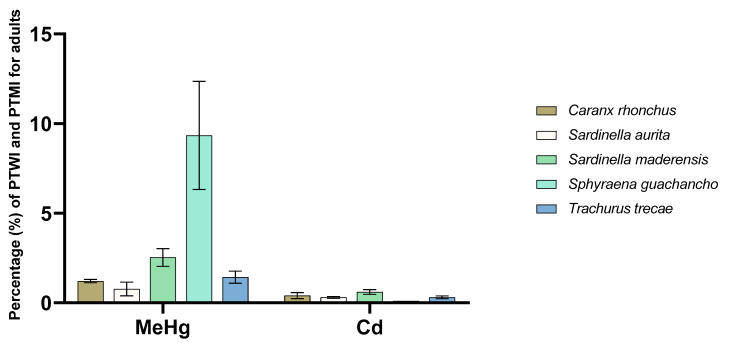
Potential consumer exposure to methylmercury (MeHg) and cadmium (Cd) for adults when a portion of 50 g fish fillet is consumed one time per week and every day for a month for MeHg and Cd, respectively. The potential consumer exposure is compared to the provisional tolerable weekly intake (PTWI) for MeHg and provisional tolerable monthly intake (PTMI) for Cd, as set by the Joint Expert Committee on Food Additives (JECFA).

**Figure 5 foods-09-00629-f005:**
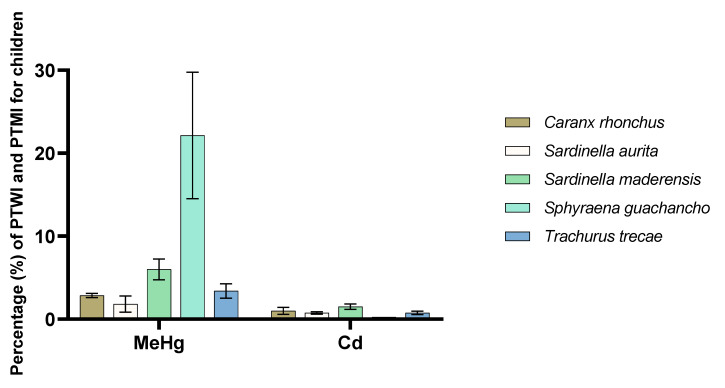
Potential consumer exposure to methylmercury (MeHg) and cadmium (Cd) for children when a portion of 25 g fish fillet is consumed one time per week and every day for a month for MeHg and Cd, respectively. The potential consumer exposure is compared to the provisional tolerable weekly intake (PTWI) for MeHg and provisional tolerable monthly intake (PTMI) for Cd, as set by the JECFA.

**Table 1 foods-09-00629-t001:** Identification details and overview of fish species sampled from Angola.

Scientific Name ^a^	Common Name ^a^	Local Name	Habitat ^a^	Weight (g) ^b^	Length (cm) ^b^	Number of Composite Samples
***Caranx rhonchus***	False scad	Chicharro amarelo	Benthopelagic	244 (210–314)	27.5 (26.5–30.5)	5
***Lagocephalus laevigatus*^c^**	Smooth puffer	Baiacu	Pelagic	734 (429–1360)	38.2 (32.5–44)	5
***Sardinella aurita***	Round sardinella	Lombuda	Pelagic	198 (176–237)	27.6 (26.5–29)	5
***Sardinella maderensis***	Madeiran sardinella	Pallheta	Pelagic	131 (116–151)	25.5 (25–27)	5
***Sphyraena guachancho***	Guachanche barracuda	Barracuda	Pelagic-neritic	509 (339–795)	46.4 (41.5–53)	5
***Trachurus trecae***	Cunene horse mackerel	Carapau do Cunene	Benthopelagic	200 (173–240)	26.1 (24.5–29)	5

^a^ Scientific name, common name, and habitat confirmed using the global species database, FishBase [33]. ^b^ Weight and length measurements are expressed as the mean values for the 25 fish and the range (minimum–maximum). Weight was measured before the cleaning of the fish. ^c^ Analytical values for this species are further presented in the Appendix A due to the species’ irrelevance for local food consumption.

**Table 2 foods-09-00629-t002:** Description of food products used for comparison to the sampled species and their edible product conversion factors ^a^.

Food	Product Description	Edible Conversion Dactor
**Chicken**	Light meat with skin, raw	1
**Beef**	Meat, moderately fat (ca. 20% fat), raw	1
**Pork**	Meat, moderately fat (ca. 20% fat), raw	1
**Milk**	Cow, whole, 3.5% fat (pasteurised or UHT)	1
**Eggs**	Chicken, raw	0.87

^a^ Products derived from the FAO/INFOODS Food Composition Table for Western Africa (2019) [28].

**Table 3 foods-09-00629-t003:** Analytical values of the proximate composition of fish species from Angola ^a^.

Species	*n* ^b^	Moisture	Protein	Lipid
		%	g/100 g	g/100 g
*Caranx rhonchus*	5	73.2 ± 0.3	23 ± 0.0	2.20 ± 0.3
*Sardinella aurita*	5	69.8 ± 0.8	21 ± 0.5	5.40 ± 2.2
*Sardinella maderensis*	5	74.2 ± 0.7	21 ± 0.4	2.33 ± 0.5
*Sphyraena guachancho*	5	75.5 ± 0.6	21 ± 0.4	2.12 ± 0.8
*Trachurus trecae*	5	70.8 ± 1.1	21 ± 0.4	6.98 ± 1.1
**Mean Content**		72.7 ± 2.3	21 ± 1.0	3.81 ± 2.3

^a^ Values are presented as means ± standard deviations (SD) of the fish species analysed in triplicate, expressed as the nutrient content per 100 g of raw, edible sample. ^b^ Number of pooled samples analysed. Each pooled sample consisted of 5 individual fish.

**Table 4 foods-09-00629-t004:** Concentrations of minerals and trace elements in fish species from Angola ^a^.

Species	*n* ^b^	Ca	Fe	I	K	Mg	Na	P	Se	Zn
		mg/100 g	mg/100 g	µg/100 g	mg/100 g	mg/100 g	mg/100 g	mg/100 g	µg/100 g	mg/100 g
*Caranx rhonchus*	5	25.8 ± 18	0.65 ± 0.01	22.0 ± 1.1	480 ± 7.1	37.4 ± 0.5	50 ± 0.8	294 ± 8.9	29.4 ± 1.7	0.47 ± 0.04
*Sardinella aurita*	5	71.2 ± 9.4	1.84 ± 0.15	23.8 ± 2.7	510 ± 10	37.6 ± 0.5	45 ± 1.6	344 ± 8.9	37.0 ± 2.2	0.52 ± 0.02
*Sardinella maderensis*	5	89.6 ± 9.9	1.52 ± 0.18	35.0 ± 4.8	528 ± 13	39.2 ± 0.8	47 ± 3.5	340 ± 7.1	50.4 ± 0.5	0.56 ± 0.07
*Sphyraena guachancho*	5	20.4 ± 3.6	0.40 ± 0.05	46.6 ± 3.6	478 ± 128	38.2 ± 0.8	45 ± 2.3	310 ± 7.1	26.2 ± 1.5	0.33 ± 0.01
*Trachurus trecae*	5	24.6 ± 7.0	0.84 ± 0.01	27.2 ± 1.3	456 ± 11	35.0 ± 0.7	65 ± 1.1	284 ± 5.5	38.2 ± 1.8	0.42 ± 0.03
**Mean Concentration**		46.3 ± 31	1.04 ± 0.56	31.0 ± 9.6	490 ± 60	37.5 ± 1.6	50 ± 7.8	314 ± 26	36.2 ± 8.7	0.46 ± 0.09

^a^ Values are presented as means ± SD for the fish species analysed in triplicate, expressed as the nutrient content per 100 g of raw, edible sample. ^b^ Number of pooled samples analysed. Each pooled sample consisted of 5 fish.

**Table 5 foods-09-00629-t005:** Concentrations of arsenic (As), cadmium (Cd), mercury (Hg), and lead (Pb) in fish from Angola (mean ± SD).

Species	*n* ^a^	As	Cd	Hg	Pb
		mg/kg w.w.	mg/kg w.w.	mg/kg w.w.	mg/kg w.w.
*Caranx rhonchus*	5	1.54 ± 0.09	0.004 ± 0.002	0.022 ± 0.002	0.006 ± 0.000
*Sardinella aurita*	5	1.78 ± 0.20	0.003 ± 0.001	0.014 ± 0.007	0.007 ± 0.000
*Sardinella maderensis*	5	2.10 ± 0.25	0.006 ± 0.001	0.046 ± 0.010	0.010 ± 0.002
*Sphyraena guachancho*	5	0.56 ± 0.11	0.001 ± 0.000	0.170 ± 0.058	0.005 ± 0.001
*Trachurus trecae*	5	2.26 ± 0.05	0.003 ± 0.001	0.026 ± 0.007	0.006 ± 0.001
**Mean Concentration**		1.65 ± 0.63	0.003 ± 0.002	0.056 ± 0.064	0.007 ± 0.002

^a^ Number of pooled samples analysed. Each pooled sample consisted of 5 fish.

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
