# Peer review of "Nutrient and Chemical Contaminant Levels in Five Marine Fish Species from Angola—The EAF-Nansen Programme"

_foods, 2020, doi:10.3390/foods9050629_

Round 1
Reviewer 1 Report
High quality nutrient data on fish species in Africa is very limited. This is in particular true when it comes to micronutrients. This paper is an important contribution in providing such data, enabling Governments to use fish products in order to improve nutrition and food security.
The paper refers to data from 2013 for the per capita fish consumption. I would recommend using more resent data for fish consumption in Angola and at Global level. The most recent data from FAO was published in 2019: FAO Yearbook, Fishery and Aquaculture Statistics 2017: http://www.fao.org/fishery/static/Yearbook/YB2017_USBcard/navigation/index_intro_e.htm
It is also important to mention that the FAO data on consumption refers to apparent consumption. For every country this is calculated as total production minus export plus import. It is not what is actually eaten. For some species where only fillets are eaten the edible part might be only 50%, or even less in some cases. For species only gutted and eaten whole, edible part might for example be 90%. This fact should be taken into account when the contribution of fish to nutrient intake is calculated and discussed. In many cases much more than only fillet are consumed. This should be more elaborated in the discussion. Small size fish eaten with skin and bones will contribute significantly more to the intake of minerals like calcium, iron and zinc. Not sure about the claim in line 620 that this is not the case. A reference would be useful to support the claim, with particular focus on micronutrients.
Contaminates included in the study did not include many of the contaminants linked to oil exploration in Angolan waters. If contamination from this sector is not a concern, some justification or references to this should be included.
In the methodology, it would be good to clarify a little more regarding the composite samples. I understand each sample represented fish fillets from five fish. It mentions that each composite sample consists of five fillets, but usually you get two fillets from each fish… Was only one fillet from each fish used for the sample?
The storage time on board at -20ºC before being transferred to the laboratory at -80ºC should be mentioned.
Reviewer 2 Report
In the present study, the authors have examined the "nutrient" and "contaminant" levels in marine species from Angola within the EAF-Nansen Programme. The results are interesting, and may provide readers and investigators with good information. However, the manuscript needs revisions to fully elucidate its findings in a research article format.
Characteristic examples are the following:
- Apart from nutrient levels, the authors examined the levels of arsenic, cadmium, mercury, and lead in fish samples from Angola. These are metal contaminants and belong to the general category of chemical contaminants. Especially mercury and lead have been named as Heavy Metals contaminants. There are other types of contaminants than metals in fish samples. Therefore, the authors need to revise in the title, abstract and throughout the manuscript the general phrase of contaminants to the more specified one of metal-contaminants.
- Introduction is too big for a research article. It contains 4 pages and a table. It should be reduced at least in 2 pages. Also the table should not be included and instead a reference can bu used or this table can be submitted as a supplementary file if it facilitates authors in their results/discussion sections.
- The authors need to rephrase the word “fat” to “lipid” content throughout the manuscript.
- In several foods and especially in fish, apart from the protein content and that of several micronutrients such as calcium, iodine, iron and zing, there are also recommended nutrient intake for women and especially for infants in essential fatty acids, such as the omega-3 fatty acids. There is no mention for these essential nutrients of fish in this study. Were they measured? If yes it would be interesting to show the levels of the omega-6/omega-3 ratio too, in these fish specie.
- In figures 1-5 the values for each species are presented as a percentage of the average RNI for each group. However, the authors have not mentioned in their statistical data management section these calculations and if normality test was performed in these data too in order to present their results as average or as median, but also they do not mention the dispersion (standard error/deviation) in these calculations, to be included as error bars in all figures 1-5.
Round 2
Reviewer 2 Report
Introduction is too big for a research article. It contains 4 pages and a table. It should be reduced in 2-3 pages. Also the previously published table in the intriduction is not really needed and the text can be revised accordingly to reference it whenever it is mentioned.
Author Response
We have now deleted the table (Table 1) from the introduction. The necessary information in this table is now provided under section 2.6 (Potential consumer exposure to metals in the methods section) where we have described the PTWI/PTMI values necessary to perform the calculations for consumer exposure. Furthermore, the maximum levels for cadmium, lead, and mercury have now only been briefly mentioned in the results where the metal concentration in fish is described (the necessary references have also been moved). Removing this table has shortened the introduction considerably. As we also removed a few sentences from the introduction, it is now under 3 pages long.